# A genome-wide meta-analysis uncovers six sequence variants conferring risk of vertigo

Astros Th. Skuladottir [1✉], Gyda Bjornsdottir[1], Muhammad Sulaman Nawaz [1,2], Hannes Petersen[2,3], Solvi Rognvaldsson [1], Kristjan Helgi Swerford Moore [1], Pall I. Olafsson[1], Sigurður H. Magnusson [1], Anna Bjornsdottir[4], Olafur A. Sveinsson[5], Gudrun R. Sigurdardottir[6], Saedis Saevarsdottir [1,2,5], Erna V. Ivarsdottir [1], Lilja Stefansdottir[1], Bjarni Gunnarsson[1], Joseph B. Muhlestein[7,8], Kirk U. Knowlton[7,8], David A. Jones[9], Lincoln D. Nadauld[9,10], Annette M. Hartmann[11], Dan Rujescu[11], Michael Strupp[12], G. Bragi Walters [1,2], Thorgeir E. Thorgeirsson[1], Ingileif Jonsdottir [1,2], Hilma Holm [1], Gudmar Thorleifsson[1], Daniel F. Gudbjartsson [1], Patrick Sulem [1], Hreinn Stefansson[1] & Kari Stefansson [1,2✉]

Vertigo is the leading symptom of vestibular disorders and a major risk factor for falls. In a genome-wide association study of vertigo ($N_{cases} = 48{,}072$, $N_{controls} = 894{,}541$), we uncovered an association with six common sequence variants in individuals of European ancestry, including missense variants in *ZNF91, OTOG, OTOGL,* and *TECTA,* and a *cis*-eQTL for *ARMC9*. The association of variants in *ZNF91, OTOGL,* and *OTOP1* was driven by an association with benign paroxysmal positional vertigo. Using previous reports of sequence variants associating with age-related hearing impairment and motion sickness, we found eight additional variants that associate with vertigo. Although disorders of the auditory and the vestibular system may co-occur, none of the six genome-wide significant vertigo variants were associated with hearing loss and only one was associated with age-related hearing impairment. Our results uncovered sequence variants associating with vertigo in a genome-wide association study and implicated genes with known roles in inner ear development, maintenance, and disease.

[1] deCODE genetics/Amgen Inc., Reykjavik, Iceland. [2] Faculty of Medicine, University of Iceland, Reykjavik, Iceland. [3] Akureyri Hospital, Akureyri, Iceland. [4] Heilsuklasinn Clinic, Reykjavik, Iceland. [5] Landspitali—The National University Hospital of Iceland, Reykjavik, Iceland. [6] Laeknasetrid Clinic, Reykjavik, Iceland. [7] Intermountain Medical Center, Intermountain Heart Institute, Salt Lake City, UT, USA. [8] University of Utah, School of Medicine, Salt Lake City, UT, USA. [9] Precision Genomics, Intermountain Healthcare, Saint George, UT, USA. [10] Stanford University, School of Medicine, Stanford, CA, USA. [11] Department of Psychiatry, Psychotherapy and Psychosomatics, Martin-Luther-University Halle-Wittenberg, Halle, Germany. [12] Department of Neurology and German Center for Vertigo and Balance Disorders, Ludwig Maximilians University, Munich, Germany. ✉email: astros.skuladottir@decode.is; kstefans@decode.is

The inner ear is complex in form. It is located in the left and right temporal bones where it frames the sensory epithelium of hearing and balance. The latter consists of the vestibular end organs, the semicircular canals, and the otolith organs connected by the vestibular nerve and nuclei to the brainstem, cerebellum, and the vestibular cortex. Together they form the vestibular system that is responsible for sensing the direction and degree of head acceleration and the pull of gravity[1]. Disturbances of these vestibular functions can be peripheral or central, resulting in vertigo, a leading symptom of various diseases and conditions, such as migraine, adverse drug effects[2], and disturbed blood pressure regulation[3]. Peripheral causes include benign paroxysmal positional vertigo (BPPV), Menière's disease, and vestibular neuritis. Central causes include brainstem and cerebellar ischemia.

Vertigo is the disturbing illusion of motion, most commonly rotational motion, of oneself or the environment. It is a major risk factor for falls and bone fractures, placing a great burden on the healthcare system[4]. Vertigo spells can develop suddenly and last for a few seconds or may be constant and last for several days, making activities of daily life more difficult. The prevalence of vertigo is 6.5%, increasing with age, and around 65% of patients are females[5].

Epidemiological studies show familial aggregation in Menière's disease[6] and higher prevalence in Caucasians than in other ethnic groups[7,8]. A genome-wide association study (GWAS) has not been previously reported for the broad phenotype of vertigo. However, a GWAS for vestibular neuritis uncovered association with four sequence variants[9].

The diagnoses of specific vestibular disorders rely on clinical criteria, as no diagnostic biomarkers are currently available. Novel technology, such as testing of vestibular evoked myogenic potentials, allows objective measurements of the vestibular end organs and thus, more accurate diagnoses[10]. Generally, vestibular compensation and habituation are recommended for those suffering from vestibular disorders[11].

Here, we describe a GWAS meta-analysis on vertigo combining data from Iceland, the UK, the US, and Finland that uncovers six associations at six different loci in genes with known roles in inner ear development, maintenance, and diseases, and evaluate their effects on the most common vestibular disorders. Additionally, we consider the relationship between age-related hearing impairment (ARHI), motion sickness and vertigo.

## Results

**GWAS meta-analysis**. We conducted a GWAS of vertigo in Iceland (30,802 cases and 278,502 controls), the UK (9715 cases and 421,332 controls), the US (1888 cases and 24,961 controls), and Finland (5667 cases and 169,746 controls). Vertigo cases were identified using the International Classification of Primary Care (ICPC-2) diagnostic code N17 in Iceland and the International Classification of Diseases (ICD-10) diagnostic code H81 in the UK, the US, and Finland. We combined 62,056,310 million variants identified through whole-genome sequencing and imputation of chip-typed individuals (see "Methods"). We tested the association between phenotype and genotype in Iceland, the UK, and the US by applying logistic regression assuming an additive model. We combined the GWAS summary results of the four datasets in a fixed-effects inverse variance model. An association was considered significant if the combined $P$ value of the datasets was below a weighted genome-wide significance threshold based on predicted functional impact of association signals[12].

We identified six associations with vertigo at six different loci (Fig. 1 and Supplementary Data 1, and Table 1), all with common variants (minor allele frequency [MAF] ≥ 5%).

The variants in *ZNF91*, *TECTA*, *OTOG*, and *OTOGL* are missense and the variants upstream of *ARMC9* and *OTOP1* are not correlated with missense variants (Fig. 2). The vertigo association at *ARMC9* co-localizes with the top *cis*-eQTL for *ARMC9* ($r^2 = 0.95$) in adipose tissue ($P = 6.1 \times 10^{-21}$, effect = 0.55 SD) in Iceland (Supplementary Fig. 1, Supplementary Data 2) and is in high linkage disequilibrium (LD; $r^2 > 0.8$) with the top *cis*-eQTL in 14 tissues in GTEx (Supplementary Data 3). The missense variant in *ZNF91* is in high LD ($r^2 > 0.8$) with the top *cis*-eQTL in *LINC01224* in brain tissue.

Using a conventional GWAS $P$ value threshold of $5 \times 10^{-8}$, we uncover three additional associations (Supplementary Data 1).

Conditional association analyses at the six loci did not reveal secondary signals.

Vertigo cases in the Icelandic dataset represent a broad phenotype as they are based on healthcare encounters in primary care using ICPC-2, in contrast to the other datasets, where cases have more specific ICD-10 diagnoses. There is evidence of heterogeneity for two of the six variants ($P$-het ≤ 0.05/ 6 = $8.3 \times 10^{-3}$, Supplementary Table 1), possibly reflecting the differences of phenotype classification in the datasets. To address this, we performed a random-effects analysis assuming that there may be different underlying true effects estimated in each dataset. We show that using a random-effects model, all of the variants remain significant given the threshold we use (Supplementary Table 1).

All of the variants are in the same direction in the four datasets, except for rs612969-G in *TECTA*, which is in the opposite direction in the Finnish dataset ($P = 0.78$, OR = 0.99; Fig. 3d). The most significant variant, rs428549-G in *ZNF91*, is genome-wide significant in the two largest datasets, Iceland ($P = 1.5 \times 10^{-9}$, OR = 1.07) and the UK ($7.8 \times 10^{-20}$, OR = 1.15). Furthermore, rs7130190-T in *OTOG* and rs10862089-T in *OTOGL* are significant in the UK ($P = 4.2 \times 10^{-8}$, OR = 1.11; $P = 7.7 \times 10^{-11}$, OR = 1.18).

We estimated the SNP heritability of vertigo in Iceland and vestibular disorders in the UK and the US using LD score regression[13]. The estimated SNP heritability in Iceland is 0.12 (95% CI 0.055−0.18) and 0.23 (95% CI 0.13−32) in the UK. The SNP heritability was not significant in the US.

We constructed a genetic risk score based on effect estimates of the six lead sequence variants from the meta-analysis, excluding the UK, and predicted into the UK dataset. The increase in variance explained was 0.26% ($\Delta R^2 = 0.0026$, $P = 1.4 \times 10^{-50}$, OR = 6.52).

We performed genetic correlation analyses using the vertigo meta-analysis and 600 GWASs reported in the UK Biobank[14]. The strongest genetic correlation was with pain traits (Supplementary Fig. 4, Supplementary Data 4).

**Gene-based genome-wide association analysis**. We conducted a gene-based genome-wide association analysis where evidence from multiple genetic variants in the same gene were combined to identify signals that are not present in a standard GWAS[15]. We tested 18,815 genes and identified 34 genes at seven loci that associate significantly with vertigo ($P ≤ 0.05/18,815 = 2.7 \times 10^{-6}$; Supplementary Fig. 5, Supplementary Table 2), which included five of the loci identified in the meta-analysis. Additional significant loci were observed at 5p13.3 and 6q25.1.

**Vertigo and related phenotypes**. Vertigo and hearing loss can co-occur in an underlying inner ear disorder, namely Menière's disease[16]. Although rare variants in *TECTA*, *OTOG*, and *OTOGL* have been associated with hearing loss (Table 1), none of the variants identified in the current meta-analysis associate with hearing loss, identified by ICD-10 codes H90 and H91

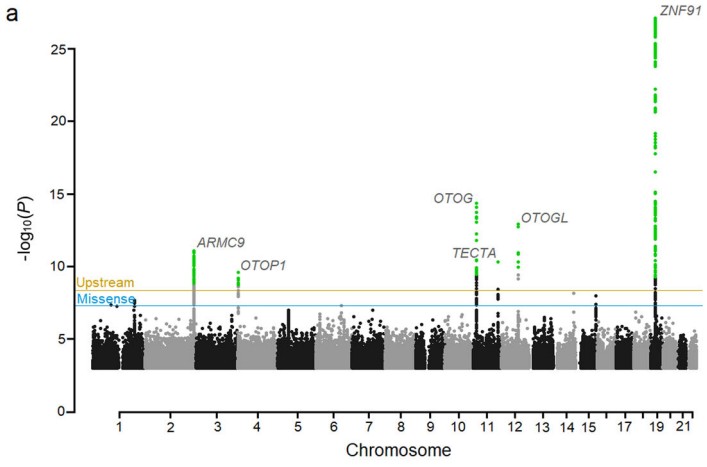

**Fig. 1 Variants reaching genome-wide significance in a meta-analysis of vertigo. a** A Manhattan plot showing six genome-wide significant loci. The horizontal lines represent the adjusted variant-class threshold (blue for missense variants [$P \leq 4.9 \times 10^{-8}$] and orange for upstream variants [$P \leq 4.4 \times 10^{-9}$]). Variants with a $P$ value below their variant-class threshold are marked in green. The $-\log_{10}P$ values ($y$-axis) are plotted for each variant against their chromosomal position ($x$-axis). Manhattan and quantile−quantile (Q-Q) plots[66] for each dataset are shown in Supplementary Figs. 2 and 3. **b** Significance and odds ratios are shown for the combined analysis (see Fig. 3 and Supplementary Data 1 for association results for each dataset). Support for gene, where OMIM entry or animal studies support the implication of the genes in vestibular functions can be seen in detail in Table 1. *MAF* minor allele frequency, *eQTL* expression quantitative trait loci, *OR* odds ratio, *CI* confidence interval.

---

**Table 1 Genes associated with vertigo, their function and previous associations.**

| Gene | Function of gene | Previous evidence in human and animal studies |
|---|---|---|
| *ZNF91* | Primate-specific KRAB zinc finger gene, rapidly evolved to repress SINE-VNTR-Alu (SVA) transposons and long interspersed nuclear element-1 (LINE-1)[56], elements that can lead to sporadic diseases via e.g. exon shuffling and alternative splicing. | — |
| *TECTA* | Encodes α-tectorin, one of the major components of the tectorial membrane, an extracellular matrix covering the neuroepithelium in the inner ear that contacts the stereocilia, a specialized sensory hair cell bundle[30]. | Autosomal dominant non-syndromic hearing impairment in humans[57] and mice[58, 59] and a recessive form of sensorineural pre-lingual non-syndromic deafness in humans[60]. |
| *ARMC9* | Encodes Armadillo repeat containing 9 which localizes to the ciliary basal body and daughter centriole and has a predicted function in ciliogenesis[22]. | Autosomal recessive Joubert syndrome and ciliopathy phenotypes in zebrafish[22]. |
| *OTOG* | Encodes the glycoprotein Otogelin, present in acellular membranes covering six sensory epithelial patches of the inner ear. Involved in anchoring otolithic membranes and cupulae to the neuroepithelia in the vestibule and organizing the fibrillary network that composes the tectorial membrane in cochlea[27]. | Autosomal recessive deafness[61] and Menière's disease[62] in humans and mice and severe imbalance in mice[27]. |
| *OTOGL* | A paralog of *OTOG*. Structural and expressional similarities suggest similar function[24, 26]. | Autosomal recessive hearing loss and sensorineural hearing loss in zebrafish[24, 26]. |
| *OTOP1* | Otopetrin 1 is a proton-selective ion channel that is required for the formation of otoconia, calcium carbonate crystals that detect linear acceleration and gravity[63]. | Unusual bilateral vestibular pathology in the absence of otoconia without hearing impairment in mice and zebrafish[64, 65]. |

(Supplementary Table 3), nor any rare variants in the six GWAS candidate genes. However, in a meta-analysis combining age-related hearing impairment (ARHI) data from Iceland and the UK ($N_{cases} = 121{,}934$, $N_{controls} = 591{,}699$)[17], rs612969-G in

*TECTA* has a small effect on ARHI ($P = 9.9 \times 10^{-5}$, OR = 1.02). Rare variants in the six GWAS candidate genes do not associate with ARHI. Motion sickness is another vertigo-related phenotype that occurs in healthy individuals due to sensory and central

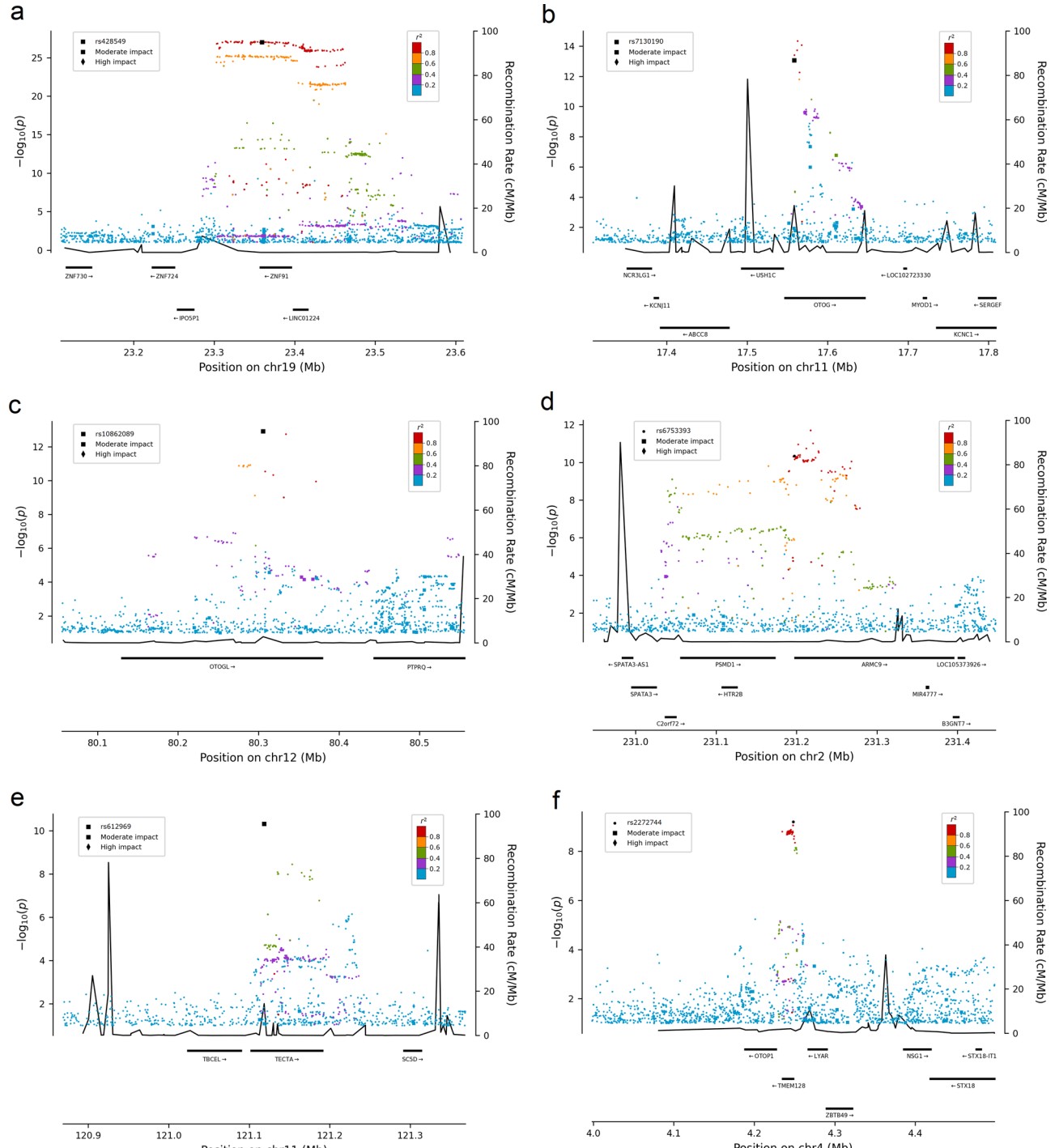

**Fig. 2 Regional plot of the loci associating with vertigo.** Regional plots representing the association with vertigo in the meta-analysis at the **a** *ZNF91* locus, **b** *OTOG* locus, **c** *OTOGL* locus, **d** *ARMC9* locus, **e** *TECTA* locus, and **f** *OTOP1* locus. Variants are colored by the degree of correlation ($r^2$) with the lead variant, which is colored black. Functional variants have a squared (moderate impact) or a diamond shape (high impact). The $-\log_{10}P$ values on the left *y*-axis (two-sided logistic regression) are plotted for each variant against their chromosomal position (*x*-axis). The right *y*-axis shows calculated recombination rates based on the Icelandic data at the chromosomal location, plotted as solid black lines.

nervous system computational conflict of postural control[18,19]. We therefore tested 50 variants reported to associate with ARHI[17] (Supplementary Data 5) and 34 variants reported to associate with motion sickness[20] (Supplementary Data 6) for association with vertigo. Two of the ARHI variants associated with vertigo (rs7525101-T, EAF = 46.0%, $P = 1.6 \times 10^{-4}$, OR = 1.03 [95% CI 1.01−1.04] and rs2242416-A, EAF = 38.2%, $P = 2.6 \times 10^{-4}$, OR = 1.03 [95% CI 1.01−1.05]) after accounting for multiple

testing ($P \leq 0.05/84 = 6.0 \times 10^{-4}$). Six reported motion sickness variants also associated with vertigo. The most significant variant is close to *LINGO2* (rs2150864-G, EAF = 33.1%, $P = 3.5 \times 10^{-6}$, OR = 1.04 [95% CI 1.02−1.06]). LINGO2 participates in neurological pathways[21]. In a weighted linear regression using the previously reported variants, we did not detect a significant correlation between the effect estimates of vertigo and ARHI ($P = 0.73$) or motion sickness ($P = 0.086$).

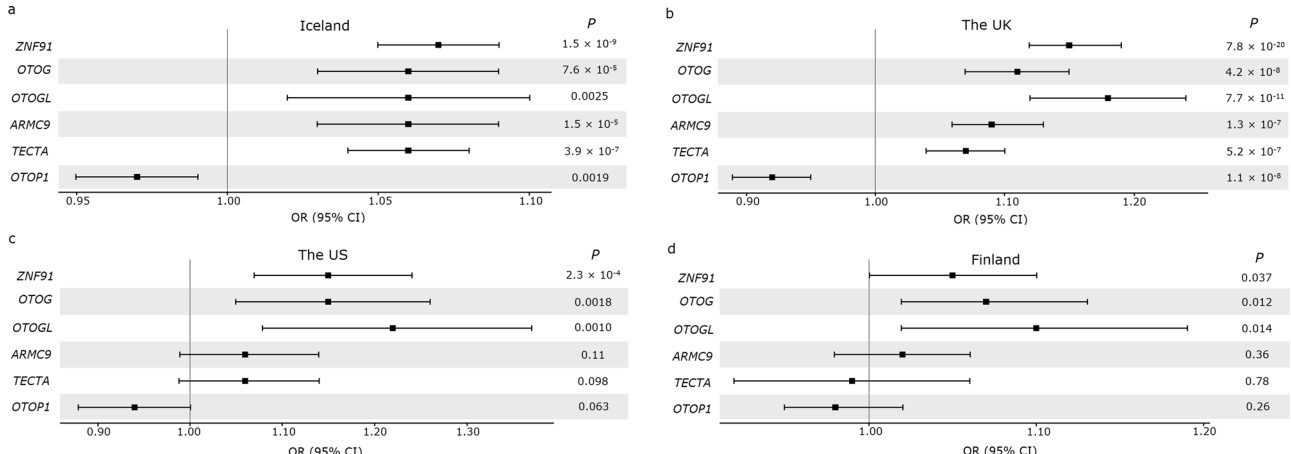

**Fig. 3 Risk comparison of the six lead sequence variants uncovered in the meta-analysis in the four datasets.** A forest plot comparing the risk of the sequence variants (represented by the gene for simplification) associating with vertigo in the meta-analysis (Supplementary Data 1), in **a** Iceland (30,802 cases and 278,502 controls), **b** the UK (9715 cases and 421,332 controls), c the US (1888 cases and 24,961 controls), and **d** Finland (5667 cases and 169,746 controls). The error bars indicate 95% CI.

**Subtypes of vertigo**. To evaluate the effect of the six variants on the most frequent peripheral vestibular disorders, we performed meta-analyses combining data from Iceland, the UK, Finland and additional small, but well-characterized, datasets from Germany. The subtypes were classified with ICD-10 codes H81.0 (Menière's disease; $N_{cases} = 3516$, $N_{controls} = 816,334$), H81.1 (BPPV; $N_{cases} = 10,947$ $N_{controls} = 848,201$), and H81.2 (vestibular neuritis; $N_{cases} = 2324$, $N_{controls} = 825,800$). There is evidence that the associations at ZNF91, OTOGL and OTOP1 are driven by BPPV after accounting for multiple testing ($P \leq 0.05/6 = 0.0083$, Supplementary Fig. 6 and Supplementary Data 7).

A previously reported GWAS of vestibular neuritis uncovered four sequence variants[9]. None of the variants were significant in our vestibular neuritis meta-analysis and the risk was significantly different between the two studies (Supplementary Data 8).

## Discussion

Vertigo is the leading symptom of various peripheral and central vestibular disorders with different underlying pathophysiologies and etiologies. For most of them, the genetic basis is largely unknown. Here, we report a GWAS meta-analysis of vertigo that uncovers six sequence variants at six loci using a weighted Bonferroni significance threshold and implicate genes coding for proteins that have a role in the biology of the inner ear. In addition, we report an association between vertigo and eight sequence variants that have previously been associated with ARHI and motion sickness. Rare variants in three of the six genes identified in the meta-analysis are reported to cause hearing loss (Table 1). However, none of the six vertigo variants reported here associate with hearing loss, although the missense variant in TECTA confers low risk of ARHI.

The inner ear is a complex, fluid-filled structure housing the peripheral sensory part of the auditory and vestibular system. The main components of the inner ear are the bony labyrinth, composed of cochlea, vestibule and three semicircular canals, and the otolith organs. The sensory epithelia of the inner ear consist of highly organized mechanosensory hair cells and non-sensory supporting cells. The basic mechanoelectrical transduction of sensory input, bending the cilia of the hair cells, is the same in the auditory and vestibular systems[1]. However, considerable variation exists in the hair cell morphology. The vestibular hair cells have a primary cilium, termed kinocilium. ARMC9 localizes to the basal bodies of primary cilia and is upregulated during ciliogenesis[22].

Bending the stereocilia toward and away from the kinocilium polarizes the cell and thus, alters the rate of nerve impulses via ciliated sensory neurons to the brainstem[23]. The variant upstream of ARMC9 that associates with vertigo (rs6753393-C, EAF = 26.6%, $P = 4.6 \times 10^{-11}$, OR = 1.06) associates with higher expression of ARMC9 ($P = 6.1 \times 10^{-21}$, effect = 0.55 SD) in adipose tissue. Based on our results, we speculate that the upstream variant rs6753393-C may disrupt the signal from hair cells to the brainstem through an increased expression of ARMC9.

The non-sensory supporting cells form the acellular membranes. The tectorial membrane along with the cupula and the otolithic membrane make up the three acellular membranes of the inner ear, where they form an intimate contact with and transmit primary stimulus to the stereocilia bundles of the hair cells[24]. The tectorial membrane is complex and consists of various proteins, one of the main proteins being α-tectorin, encoded by TECTA. We detected a missense variant in TECTA (Arg371Gly) that associates with vertigo (rs612969-G, EAF = 44.1%, $P = 4.9 \times 10^{-11}$, OR = 1.06). In the mouse cochlea, Tecta mRNA levels are highest in early postnatal stages, and dramatically decrease in adult stages suggesting an important function of Tecta during tectorial membrane morphogenesis[25]. Analyzing Icelandic eQTL data, we saw that TECTA mRNA is present in the adipose tissues of adults. However, the missense variant does not associate with the expression of the gene. This does not exclude the possibility that the variant affects expression of TECTA in other tissues in adult stages or the expression of TECTA in earlier life stages as in the mouse cochlea.

It has been suggested that OTOG and OTOGL play a similar role in the inner ear because of their similar structure and expression patterns[24] and mutations in these two genes have been reported to associate with similar hearing impairment phenotypes in humans[26] and deafness and severe imbalance in mice[27]. Here, we report an association between vertigo in humans and a missense mutation in OTOG (Thr375Ser, rs7130190-T, EAF = 16.2%, $P = 9.0 \times 10^{-14}$, OR = 1.08) and OTOGL (Gln1102His, rs10862089-T, EAF = 7.50%, $P = 1.2 \times 10^{-13}$, OR = 1.11). The high transcriptional levels of Otogl and Otog in embryonic stages of mice are downregulated in adult stages. However, the protein product of both genes is prominently present in the acellular structures of the inner ear[24,28]. This may suggest an important role in the production of the inner ear structure and a lower gene activity for the maintenance of the otolithic membranes and cupula in later stages[24]. Further, otogelin is almost

undetectable in adult cochlear cells, indicating that it is a long-lasting protein in the tectorial membrane[28].

Embedded in the otolithic membrane are calcium carbonate crystals, termed otoconia. During movement, the weight of these crystals shifts the otolithic membrane, deflecting the hair cells[1]. The vertigo association with the sequence variants at *ZNF91*, *OTOP1*, and *OTOGL* appears to be driven by their risk of BPPV. Otopetrin-1, encoded by *OTOP1*, is the most prominent protein in otoconia formation. The importance of otoconia is demonstrated in the elderly, where otoconia degenerate, and in BPPV, which may occur due to displacement of otoconia. In this study, we highlight *OTOP1* as a candidate gene because of its obvious functional role in the otoconia formation. However, rs2272744 is closer to another gene, *TMEM128*, the function of which is poorly understood.

Vertigo and hearing problems often co-occur[29–32]. None of the variants identified in the meta-analysis associate with hearing loss and only one has a modest effect on ARHI (rs612969-G in *TECTA*; $P = 9.9 \times 10^{-5}$, OR = 1.02). Furthermore, we do not observe a consistent trend between the reported effects of variants associated with ARHI and their effects on vertigo ($P = 0.73$, Supplementary Data 5), which may suggest a different pathology between the disorders of the auditory and vestibular system.

One clear limitation of this study is the absence of an inner ear tissue to study the RNA expression of the genes associated with vertigo. To mitigate this deficiency, we use the public database GTEx and RNA sequencing from Icelandic blood and adipose tissue. Another limitation is the absence of a replication dataset. However, using a large dataset such as the Icelandic dataset and combining it in a meta-analysis with datasets from three different populations has allowed us to identify a substantial dataset of 48,072 cases and 894,541 controls, a number in considerable excess of many published GWASs. The statistical power conferred by using a dataset of this size has allowed us to discover multiple genome-wide signals, even after the application of stringent QC criteria on both genotyped and imputed variants and a weighted significance threshold based on annotation.

Vertigo is common and places a great burden on the healthcare system. Our study unveiled genome-wide significant associations with vertigo and revealed sequence variants in genes that may contribute to the pathogenesis of diseases in either the auditory system or the vestibular system. Thus, the results further our understanding of the biological underpinnings of the vestibular and auditory systems. Additional functional studies exploring the role of these genes in different developmental stages of the inner ear are needed to help disentangle the role of the vertigo-associated variants.

## Methods

**Ethics statement.** All Icelandic data were collected by studies approved by the National Bioethics Committee (NBC; VSN-19-158; VSNb2019090003/03.01) following review by the Icelandic Data Protection Authority. Participants donated blood or buccal samples after signing a broad informed consent allowing the use of their samples and data in all projects at deCODE genetics approved by the NBC. All personal identifiers of the participants' data were encrypted by a third-party system, approved and monitored by the Icelandic Data Protection Authority.

The UK Biobank data were obtained under application number 24898. Phenotype and genotype data were collected following an informed consent obtained from all participants. The North West Research Ethics Committee reviewed and approved UK Biobank's scientific protocol and operational procedures (REC Reference Number: 06/MRE08/65).

The US participants were recruited by HerediGene: Population study, a large-scale collaboration between Intermountain Healthcare, deCODE genetics, and Amgen, Inc. The Intermountain Healthcare Institutional Review Board approved this study, and all participants provided written informed consent prior to enrollment.

The FinnGen database consists of samples collected from the Finnish biobanks and phenotype data collected at the national health registers. Participants are volunteers that have provided a written informed consent. The Coordinating Ethics

Committee of the Helsinki and Uusimaa Hospital District evaluated and approved the FinnGen research project. The project complies with existing legislation (in particular the Biobank Law and the Personal Data Act). The official data controller of the study is the University of Helsinki.

The study of German BPPV cases was approved by the ethics committee of the Ludwig-Maximilian University in Munich and was carried out in accordance with the Declaration of Helsinki. All participants signed an informed consent.

**Sample description.** In Iceland, we identified patients with ICPC-2 code N17 (representing a healthcare encounter due to vertigo/dizziness) in the Registry of Primary Health Care Contacts and the Registry of Contacts with Medical Specialists in Private Practice. Additionally, we searched for vertigo subcodes classified by ICD-10 codes H81.0−H81.4. Records covered the years from 1997 to 2019.

The UK Biobank study is a large prospective cohort study of ~500,000 individuals in the age range of 40−69 from across the UK. Extensive phenotype and genotype data have been collected for participants, including ICD diagnosis codes. Over 80% of the UK cases (8.422/9.715 = 86.7%) used in the meta-analysis come from General Practice clinical recent records (Field ID 42040), where vertigo diagnostic codes were translated from Read Codes Version 2 and Read Codes Clinical Terms Version 3 to ICD-10 code H81 and subcodes H81.0−H81.4 (Supplementary Table 4). The rest of the vertigo cases were identified by searching for cases with ICD-10 code H81 from UK hospital diagnoses (1.293/9.715 = 13.3%; Field ID 41270 and 41271).

Participants in the HerediGene: Population study are voluntary US residents over the age of 18 years. Subjects with ICD-10 code H81 were identified from medical records.

The phenotype data from the FinnGen study were produced from several national health registries. All vertigo cases were diagnosed by a physician and categorized using ICD-10 codes H81.0, H81.1, H81.2, H81.3, H81.4, H81.8, H81.9, ICD-9 codes 386.0, 386.1, 386.8, 386.9, and ICD-8 code 38599. The summary statistics for available phenotypes, including vertigo, were imported on November 30, 2020 from a source available to consortium partners (version 4; http://r4.finngen.fi).

In total, we had 48,072 vertigo cases (30,802 from Iceland, 9715 from the UK, 5667 from Finland, and 1888 from the US) and 894,541 controls (278,502 from Iceland, 421,332 from the UK, 169,746 from Finland, and 24,961 from the US) in the vertigo meta-analysis. All participants were genotypically verified as being of white origin.

BPPV cases (N = 335) of European descent were recruited by the German Center for Vertigo and Balance disorders (Munich, Germany). Detailed medical histories of the participants and their first-degree relatives were assessed using structured interviews. The controls (N = 2609) were healthy volunteers of German descent, selected from PAGES (Phenomics and Genomics Sample) after excluding participants with any self-reported neurological or psychiatric history. Central nervous system impairment was ruled out using an orientating neurological examination[9].

**Genotyping and imputation.** The preparation of samples and the whole-genome sequencing (WGS) of Icelanders was performed at deCODE in Iceland[33,34]. Over 34.1 million high-quality sequence variants were identified through WGS of 61,205 Icelanders using GAIIx, HiSeq, HiSeqX, and NovaSeq Illumina technology to a mean depth of at least ×17.8. Single nucleotide polymorphisms (SNPs) and insertions/deletions (indels) were identified and their genotypes called using joint calling with Graphtyper[35]. Additionally, over 155,250 Icelanders (including all sequenced Icelanders) have been genotyped using various Illumina SNP chips and phased using long-range phasing[36], which allows for improving genotype calls using the information about haplotype sharing. Subsequently, genealogic information was used to impute sequence variants into the chip-typed Icelanders and their relatives[37] to increase the sample size and power for association analysis.

The UK Biobank samples were genotyped with a custom-made Affymetrix chip, UK BiLEVE Axiom in the first 50,000 individuals[38], and the Affymetrix UK Biobank Axiom array[39] in the remaining participants. The samples from the US (Intermountain dataset) were WGS using NovaSeq Illumina technology (N = 8288) and genotyped using Illumina Global Screening Array chips (N = 28,279). Samples were filtered on 98% variant yield in both the UK and the US dataset and any duplicates removed. To identify outliers of significant non-European ancestry, we ran supervised ADMIXTURE v1.23[40] using 1000G populations CEU, CHB, ITU, PEL, and YRI[41] as training samples. We prepared training data for ancestry analysis by removing long-range LD regions[42] and second-degree relatives identified by KING 2.2.5 --kinship[43], and then removed ancestry outliers (especially PEL individuals with European ancestry) identified using SMARTPCA[44], unsupervised ADMIXTURE, and leave-one-out supervised ADMIXTURE. We then restricted training data to markers present in the Intermountain dataset and LD-pruned using PLINK v1.90b6.15[45] --indep-pairwise 200 25 0.4. We then defined a stricter ancestry subset using a principal component analysis and UMAP[46] and removed variants showing a deviation in Hardy−Weinberg equilibrium ($P < 1.0 \times 10^{-8}$) in this subset without excessive heterozygosity and/or excessive mismatches to available sequence data. Over 93 million high-quality sequence variants and indels to a mean depth of at least ×20 were identified in the UK datasets using Graphtyper[35] and over 245 million high-

quality sequence variants and indels in the US dataset. Quality-controlled chip genotype data were phased using Shapeit 4[47]. A phased haplotype reference panel was prepared from the sequence variants, where at least 50% of the samples had a GQ score > 0, using in-house tools and the long-range phased chip genotype data. We used the same methods as described above for the Icelandic data, to impute the genotypes from the haplotype reference panel into the phased chip data.

A custom-made FinnGen ThermoFisher Axiom array (>650,000 SNPs) was used to genotype ~177,000 FinnGen samples at ThermoFisher genotyping service facility in San Diego. Genotype calls were made with AxiomGT1 algorithm (https://finngen.gitbook.io/documentation/methods/genotype-imputation). Individuals with ambiguous gender, high genotype missingness (>5%), excess heterozygosity (±4 SD), and non-Finnish ancestry were excluded. Variants with high missingness (>2%), low Hardy−Weinberg equilibrium ($<1 \times 10^{-6}$), and minor allele count (<3) were excluded. Imputation was performed using the Finnish population specific and high coverage (×25−30). WGS backbone and the population-specific SISu v3 imputation reference panel with Beagle 4.1. More than 16 million variants have been imputed.

The genotyping, quality control, and imputation of the German dataset has been described in detail elsewhere[9]. In short, the genotyping of the German dataset was performed on different platforms, imputed in seven batches and combined into one large dataset. The quality control and imputation of three batches were performed in the framework of a schizophrenia meta-analysis conducted by the Psychiatric Genomics Consortium (PGC). The other four batches were processed using the same protocol used by the PGC.

**Association analysis**. We applied logistic regression assuming an additive model using the Icelandic, UK, and US data and tested for association between sequence variants and vertigo using software developed at deCODE[33]. Association results from FinnGen were imported (version 4; http://r4.finngen.fi). We used LD score regression to account for distribution inflation due to cryptic relatedness and population stratification in the Icelandic, UK, and US dataset[13].

In the Icelandic association analysis, we adjusted for sex, county of origin, current age or age at death (first- and second-order terms included), blood sample availability for the individual, and an indicator function for the overlap of the lifetime of the individual with the time span of phenotype collection[37]. GWAS was run on a subset of UKB individuals of inferred British–Irish genetic ancestry, which was defined by running UMAP[46] on the 40 genetic principal components provided by UKB using the R package umap on default settings, and circumscribing a distinct cluster comprised almost entirely of self-identified White British and Irish[48]. We adjusted for sex, age, and the first 20 principal components (Supplementary Fig. 7) to adjust for population stratification[49]. In the US association analysis, samples assigned <93% CEU ancestry were excluded. We adjusted for sex, age, and the first 20 principal components (Supplementary Fig. 8). The Finngen association analysis was adjusted for sex, age, the genotyping batch, and the first ten principal components.

We combined vertigo GWAS summary results from Iceland, the UK, Finland, and the US using a fixed-effects inverse variance method[50] based on effect estimates and standard errors in which the study groups were assumed to have a common OR but allowed to have different population frequencies for alleles and genotypes. The total number of variants included in the meta-analysis that had imputation information above 0.8 and MAF > 0.01% was 62,056,310 (22,677,509 in Iceland, 38,838,789 in the UK, 28,369,981 in the US, and 13,990,237 in Finland). We estimated the genome-wide significance threshold using a weighted Bonferroni adjustment that controls for the family-wise error rate[12]. Sequence variants were mapped to NCBI Build38 and matched on position and alleles to harmonize the four datasets. Variants were weighted based on predicted functional impact: $P \leq 2.40 \times 10^{-7}$ for high-impact variants; $P \leq 4.90 \times 10^{-8}$ for moderate-impact variants such as missense variants; $P \leq 4.40 \times 10^{-9}$ for low-impact variants such as upstream variants; $P \leq 2.20 \times 10^{-9}$ for intronic and intergenic variants in DNase I hypersensitivity sites (DHS); $P \leq 7.40 \times 10^{-10}$ for intronic and intergenic variants in other non-DHS sites.

In a random-effects method, a likelihood ratio test was performed in all genome-wide associations to test the heterogeneity of the effect estimate in the four datasets; the null hypothesis is that the effects are the same in all datasets and the alternative hypothesis is that the effects differ between datasets.

Conditional association analysis was performed on (a) the GWASs from Iceland, the UK, and the US using true imputed genotypes of participants where the adjusted P values were combined for all three datasets to identify the most likely causal variant at each locus and to identify any secondary signals, and (b) adipose tissue eQTL data from Iceland for rs6753393, adjusting for all variants in high LD ($r^2 > 0.8$) and vice versa.

Association analyses of the six variants in the German dataset were conducted with PLINK[51] using a logistic regression under an additive model, adjusting for age, sex, and the first two principal components.

**Genetic risk score**. A genetic risk score was constructed based on the effect estimates of the six lead sequence variants from the meta-analysis, excluding the UK. The risk score was then used to predict into the UK dataset. The increase in variance explained was estimated using a logistic regression model where the full model of the genetic risk score was compared to the null model, which adjusts for sex, year of birth to the third power, an interaction of these two covariates, and the first 40 principal components.

**Genetic correlation analysis**. Genetic correlation analyses between the vertigo meta-analysis and 600 published GWAS traits ($P \leq 8.3 \times 10^{-5}$) from the UK Biobank[14] with effective sample size over 5000 were performed using LD score regression[13,52], which suggests the minimal effective sample size of 5000 for each trait to get unbiased estimates of genetic correlation and heritability. Since participants in the published GWAS studies are of Caucasian ancestry, we used pre-computed LD scores from a 1000 genome panel with $r^2$ from HapMap3, excluding HLA region. The HLA region was excluded for its genetic complexity and association with a wide number of traits. The default parameters of the LD score regression were used to compute the genetic correlation and heritability estimates.

**Gene-based genome-wide association analysis**. MAGMA is a tool for gene and gene-set analysis of GWAS genotype data that uses multiple regression to incorporate LD between markers and to detect multi-marker effects[15]. SNP-based P values of 6,516,419 SNPs from the meta-analysis were combined into gene-based P values and used as input for the MAGMA software version 1.08. Gene-based P values were generated for 18,815 genes and the significance threshold was based on number of tests ($P \leq 0.05/18,815 = 2.7 \times 10^{-6}$). The 1000 Genomes reference panel was used to control for LD. Common SNPs within ±200 kb of the respective genes were included in the analysis. The NCBI Gene database was used to define the genomic intervals. No up- or downstream variants were used in the analysis.

**Transcriptomics**. RNA sequencing was performed on whole blood from 13,175 Icelanders and on subcutaneous adipose tissue from 750 Icelanders, described in detail elsewhere[53]. Gene expression was computed based on personalized transcript abundances using kallisto[54]. Association between sequence variants and gene expression (cis-eQTL) was estimated using a generalized linear regression, assuming additive genetic effect and quantile normalized gene expression estimates, adjusting for measurements of sequencing artifacts, demographic variables, blood composition, and hidden covariates[55].

**URLs**. https://gtexportal.org/home/, https://www.omim.org/

**Reporting summary**. Further information on research design is available in the Nature Research Reporting Summary linked to this article.

## Data availability
The GWAS summary statistics for vertigo are available at https://www.decode.com/summarydata/. Other data generated or analyzed during this study are included in the manuscript and the Supplements. Source data underlying main figures are provided in Supplementary Data 1.

## Code availability
We used the following publicly available software to analyze the data: GraphTyper is available at https://github.com/DecodeGenetics/graphtyper (v2.0-beta, GNU GPLv3 license). Svimmer, the structural variant merging software is available at https://github.com/DecodeGenetics/svimmer (v0.1, GNU GPLv3 license). ADMIXTURE is available at http://dalexander.github.io/admixture/download.html. KING (v. 2.2.5–kinship) is available at https://www.kingrelatedness.com/Download.shtml. SMARTPCA is available at http://rd.plos.org/david_reich_laboratory. PLINK (v1.90b6.15) is available at https://www.cog-genomics.org/plink2/. UMAP is available at https://github.com/lmcinnes/umap. SHAPEIT4 is available at https://odelaneau.github.io/shapeit4/. qqman is available at https://github.com/stephenturner/qqman. MAGMA is available at http://ctglab.nl/software/magma.

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

## Acknowledgements

We thank the participants in this study for their valuable contribution to research. We also thank our colleagues at deCODE who contributed to genotyping and analysis of the WGS data. This research was conducted using the UK Biobank Resource (application number 24898). We acknowledge Stacey Knight, Tyler Barker, Jeffrey L. Anderson, and John F. Carlquist for their contribution to the HerediGene: Population study. We acknowledge the participants and investigators of the FinnGen study. The financial

support from the European Commission to the NeuroPain project (FP7#HEALTH-2013-602891-2) and painFACT project (H2020-2020-848099), and the National Institutes of Health (R01DE022905) is acknowledged.

## Author contributions

A.T.S., G.B., H.S., and K.S. designed the study. A.T.S., G.B., M.S.N., S.R., S.H.M., S.S., B.G., L.S., A.M.H., H.H., G.T., D.F.G, P.S., H.S., and K.S. analyzed the data and interpreted the results. Data collection and subject ascertainment and recruitment was carried out by G.B., A.B., O.A.S., G.R.S., J.B.M., K.U.K., D.A.J., L.D.N., A.M.H., D.R., M.S., G.B.W., and K.S. A.T.S. drafted the manuscript with input and comments from G.B., M.S.N., K.H.S.M., P.I.O., H.P., S.S., E.V.I., M.S., G.B.W., T.E.T., I.J., H.H., G.T., D.F.G., P.S., H.S., and K.S. All authors contributed to the final version of the paper.

## Competing interests

The authors declare the following competing interests: A.T.S., G.B., M.S.N., S.R., S.H.M., E.V.I., B.G., L.S., G.T, S.S., K.H.S.M., P.I.O., G.B.W., T.E.T., H.H., P.S., D.F.G., I.J., H.S., and K.S. are employees of deCODE genetics/Amgen, Inc. M.S. is a joint chief editor of the *Journal of Neurology*, chief editor of *Frontiers of Neuro-otology* and section editor of *F1000*. He has received speaker's honoraria from Abbott, Auris Medical, Biogen, Eisai, Grünenthal, GSK, Henning Pharma, Interacoustics, J&J, MSD, Otometrics, Pierre-Fabre, TEVA, and UCB. He is a shareholder and investor of IntraBio. He distributes "M-glasses" and the "Positional vertigo App". He acts as a consultant for Abbott, Auris Medical, Heel, IntraBio and Sensorion. The remaining authors declare no competing interests.
