## [Peer Review File · Communications Biology]

Reviewers' comments:

Reviewer #1 (Remarks to the Author):

Astros Th Skuladottir et al. conducted a genome-wide association study on 48,072 vertigo cases and 894,541 controls from four populations. They identified six common variants associated with the vertigo, including missense variants in ZNF91, OTOG, OTOGL, andTECTA, and a cis-eQTL for ARMC9.

This study is a large-scale association study consist of four independent populations, and the statistic is reasonable and well performed. However, there are a few points that need further clarification.

Major comments:

1. The plot of principal components analysis, and the quantile-quantile plots for each population and meta-analysis should be provided.
2. The detail information of eQTL results of six index SNPs (especially non-coding variants rs6753393 and rs2272744) should be provided. Firstly, the authors performed analysis using RNA sequencing data of whole blood from 13,175 Icelanders and on subcutaneous adipose tissue from 750 Icelanders, as described in Methods, but they only reported the eQTL of rs6753393 in adipose tissue in Results. Moreover, a GTEx URL was presented in the article while no eQTL results or discussion about the GTEx data. It would be more convincible if the authors confirmed the eQTL associations in public databased such as the GTEx, especially using data from brain tissues.
3. It seems the colocalization of GWAS association and cis-eQTL association for ARMC9 was based on the LD r^2 between the index SNP rs6753393 and top cis-eQTL SNP. A more standard approach would be to perform colocalization analysis using specifical statistical methods embedded in Coloc/SMR tools.
4. It is confused about the fine-mapping gene of rs2272744. According to HaploReg (<http://archive.broadinstitute.org/mammals/haploreg/haploreg.php>) and the regional plots in Supplementary Figure 1, rs2272744 is located 182 bp upstream to the transcription start site of TMEM128, while far away from the OTOG1 (>20 kb). Moreover, the GTEx v8 data show that the top eQTL hit of rs2272744 is TMEM128 in Cerebellum tissues of brain.

Minor comments:

1. Some elements in regional plots in Supplementary Figure 1 and 2 were not well annotated. For example, orange should be annotated $0.6 < r^2 \leq 0.8$.
2. It is better to describe the quality control steps of genotype data of four populations more clearly, such as the Hardy-Weinberg equilibrium and call rate.
3. None of the variants identified in previously reported GWAS of vestibular neuritis were significant in the meta-analysis of this study. How about the association results in four independent population?
4. Some mistakes. For example, the line 7 of the first paragraph in "Subtypes of vertigo" section, P may be 0.0083 and the Supplementary Fig. 6 dose not exist.

Reviewer #2 (Remarks to the Author):

This is for the most part a straightforward GWAS paper to read and understand, and I enjoyed it, although some details were lacking and organizational structure could be improved in the methods section, that should be addressed. It is the first paper to be written up on vertigo.

There is no replication cohort, so a greater assessment of consistency amongst the cohorts (i.e., were all effects in the same direction) would be greatly appreciated, perhaps with a better table of it or forest plot or something in the main results, and description of this in the text, rather than having to go to the supplementary materials to assess this. For example, one variant appears to be genome-wide significant in the two largest cohorts.

In the methods, the details given for each of the cohorts needs to be harmonized better. Some details are left out for some cohorts and not others. E.g., what software was used for the UKB? This may be what gives the methods section a disorganized feel.

How was the conditional analysis done? Using individual level data in every cohort, or?

Why weren't other groups in UKB assessed? What were the number of cases in those groups?

How were cryptic relatives adjusted for? It is explicitly addressed for the Icelandic data, but not mentioned for the other cohorts, e.g., no details for the UK Biobank which has a substantial amount of known relatives.

The data availability should cover all cohorts and be explicit as to what data is available.

Reviewer #3 (Remarks to the Author):

AT Skuladottir and colleagues performed the first GWAS for vertigo in a large vertigo case control study (48K cases, 894.5K controls) of European ancestry and identified 6 significant loci. This is a well-executed and important study in the field.

MAJOR COMMENTS

1-The authors may consider calculating SNP-based / chip heritability for vertigo in the dataset.

2-In addition to conventional SNP-based GWAS, the authors may complete a genome-wide gene-based association study to identify additional loci of interest (Hägg et al., Hum Mol Genet 2015).

3-The authors may complete a genome-wide rare / common CNV association study for vertigo (Wheeler et al., Nat Genet 2013).

4-The authors used a weighted genome-wide significance threshold based on predicted functional impact of association signals in the GWAS. They may provide a brief summary of the results for conventional GWAS for vertigo without weighted adjustment for sequence variants (equal prior probability for all variants) as a comparison.

5-'The vertigo association at ARMC9 co-localizes with the top cis-eQTL for ARMC9 ($r^2 = 0.95$) in adipose tissue ($P = 3.3 \times 10^{-20}$, effect = 0.56 SD) in Iceland'. Adipocyte is an excellent target tissue for obesity, but I am not so sure for vertigo. It may be relevant to show additional cis-eQTL association data in diverse tissues (GTEx) for the SNPs.

6-The authors may consider adding GWAS gene enrichment studies for molecular / gene function pathways and expression in specific tissue.

7-The authors may estimate the % of variance for the trait explained by the 6 variants in an independent dataset.

8-The authors may check whether the six GWAS lead SNPs or SNPs in strong linkage disequilibrium are associated with other traits / diseases in the GWAS catalogue (Kaur et al., Obes Rev 2019).

9-'None of the variants identified in the meta-analysis associate with hearing loss and only one has a modest effect on ARHI (rs612969-G in TECTA; $P = 9.9 \times 10^{-5}$, OR = 1.02), which may indicate a different pathology, in part, in these two systems'. I do not agree at all with this statement. This is inconsistent with the fact that rare variants in three of the six genes identified in the meta-analysis are reported to cause hearing loss. Statistical power may explain the divergent findings for SNPs. Did you consider screening the three remaining GWAS candidate genes for rare variants associated with hearing loss?

10-The authors may consider adding a strengths / limitations section in the discussion section.

MINOR COMMENTS

1-Please mention in the title and abstract that the study has been performed in participants of European ancestry.

2-Please define what the abbreviation BBPV means.

3-Figure 1: please provide more explanation for the 'support for gene' section

Response to reviewers' comments

We appreciate the time and effort the reviewers dedicated to, truly, improving the manuscript with their insightful and constructive comments. Most of the suggestions have been incorporated into the manuscript and Supplementary information using track changes. Below are our detailed responses.

Reviewer 1

Astros Th Skuladottir et al. conducted a genome-wide association study on 48,072 vertigo cases and 894,541 controls from four populations. They identified six common variants associated with the vertigo, including missense variants in *ZNF91*, *OTOG*, *OTOGL*, and *TECTA*, and a cis-eQTL for *ARMC9*. This study is a large-scale association study consist of four independent populations, and the statistic is reasonable and well performed. However, there are a few points that need further clarification.

Major comments:

- 1.1 The plot of principal components analysis, and the quantile-quantile plots for each population and meta-analysis should be provided.

Response (1.1)

The UK Biobank is a well-established public dataset and the principal components have been previously estimated¹. The first 20 principal components used in this study were defined as a subset from the 40 principal components provided by the UK biobank¹. The first 20 principal components in the US dataset were calculated in the same manner as for the UK dataset. We have added plots of the principal component analysis for the UK and US datasets (Supplementary Fig. 8 and 9). The FinnGen dataset is a public-private source combining genotype data from Finnish biobanks and digital health records from Finnish health registries. The principal component analysis (PCA) is described in detail on their website (<https://finngen.gitbook.io/documentation/methods/phewas/quality-checks>). We have also added quantile-quantile plots for the meta-analysis and the individual datasets (Supplementary Fig. 4).

- 1.2 The detail information of eQTL results of six index SNPs (especially non-coding variants rs6753393 and rs2272744) should be provided. Firstly, the authors performed analysis using RNA sequencing data of whole blood from 13,175 Icelanders and on subcutaneous adipose tissue from 750 Icelanders, as described in Methods, but they only reported the eQTL of rs6753393 in adipose tissue in Results. Moreover, a GTEx URL was presented in the article while no eQTL results or discussion about the GTEx data. It would be more convincible if the authors confirmed the eQTL associations in public databased such as the GTEx, especially using data from brain tissues.

Response (1.2)

The reviewer is correct. We have added a detailed eQTL results for the lead variants using blood and adipose eQTL data from Iceland and 49 tissues from GTEx (<https://www.gtexportal.org/home/>) to Supplementary Data 3. The eQTL results for rs6753393 in adipose tissue were the only significant eQTL results in the Icelandic dataset, where the reported variant associated with expression of the candidate gene. We have added that rs6753393 is in high LD ($r^2 > 0.8$) with the top cis-eQTL in 14

tissues in GTEx (page 4). The missense variant in *ZNF91*, rs428549, is in high LD with the top eQTL in another gene, *LINC01224*, in brain tissue.

- 1.3** It seems the colocalization of GWAS association and cis-eQTL association for *ARMC9* was based on the LD r^2 between the index SNP rs6753393 and top cis-eQTL SNP. A more standard approach would be to perform colocalization analysis using specific statistical methods embedded in Coloc/SMR tools.

Response (1.3)

Like COJO, COLOC and SMR approximate the variance-covariance matrix between association statistics with the LD information from a reference panel. Instead of using approximate methods, we used true imputed genotypes of participants and performed conditional analyses on a) the GWASs from Iceland, the UK, and the US, and b) adipose tissue eQTL data from Iceland for rs6753393, adjusting for all variants in high LD ($r^2 > 0.8$) and vice versa (see page 20 in Methods). Based on LD information from Iceland, 51 markers correlate ($r^2 > 0.8$) with rs6753393. We have added the results from the conditional analyses to Supplementary Data 2.

- 1.4** It is confused about the fine-mapping gene of rs2272744. According to HaploReg (<http://archive.broadinstitute.org/mammals/haploreg/haploreg.php>) and the regional plots in Supplementary Figure 1, rs2272744 is located 182 bp upstream to the transcription start site of *TMEM128*, while far away from the *OTOP1* (>20 kb). Moreover, the GTEx v8 data show that the top eQTL hit of rs2272744 is *TMEM128* in Cerebellum tissues of brain.

Response (1.4)

The variant rs2272744 maps to chr4:4248406 in GRCh38. According to Ensembl, *OTOP1* maps to chr4:4188726-4226929 (ENSG00000163982) and *TMEM128* maps to chr4:4235542-4248223 (ENSG00000132406). Thus, rs2272744 is closer to *TMEM128* than *OTOP1*, as the reviewer correctly pointed out. However, we highlighted the obvious functional candidate, *OTOP1*, which encodes Otopetrin-1, the most prominent protein in otoconia formation.

We have added to the Discussion (page 12)

“In this study, we highlight *OTOP1* as a candidate gene because of its obvious functional role in the otoconia formation. However, rs2272744 is closer to another gene, *TMEM128*, the function of which is poorly understood.”

and we have also added eQTL data from GTEx for *OTOP1* and *TMEM128* to Supplementary Data 3.

Minor comments:

- 1.5** Some elements in regional plots in Supplementary Figure 1 and 2 were not well annotated. For example, orange should be annotated $0.6 < r^2 \leq 0.8$.

Response (1.5)

The annotation has been simplified (see Supplementary Fig. 1 and 2).

- 1.6** It is better to describe the quality control steps of genotype data of four populations more clearly, such as the Hardy-Weinberg equilibrium and call rate.

Response (1.6)

The genotype and imputation sections have been reconstructed and improved for the UK and US datasets (page 17). We have also added information for the Finnish dataset (page 18).

- 1.7** None of the variants identified in previously reported GWAS of vestibular neuritis were significant in the meta-analysis of this study. How about the association results in four independent population?

Response (1.7)

We have expanded Supplementary Table 5 (now Supplementary Data 6) to show results for each of the datasets included in the vestibular neuritis meta-analysis. None of the variants are significant in the independent populations.

- 1.8** Some mistakes. For example, the line 7 of the first paragraph in “Subtypes of vertigo” section, P may be 0.0083 and the Supplementary Fig. 6 does not exist.

Response (1.8)

We thank the reviewer for pointing this out. The P has been edited from 0.0.83 to 0.0083 and after incorporating changes using the suggestions from the reviewers, “Supplementary Fig. 6” is now “Supplementary Fig. 7” (page 10).

Reviewer 2

This is for the most part a straightforward GWAS paper to read and understand, and I enjoyed it, although some details were lacking and organizational structure could be improved in the methods section, that should be addressed. It is the first paper to be written up on vertigo.

- 2.1** There is no replication cohort, so a greater assessment of consistency amongst the cohorts (i.e., were all effects in the same direction) would be greatly appreciated, perhaps with a better table of it or forest plot or something in the main results, and description of this in the text, rather than having to go to the supplementary materials to assess this. For example, one variant appears to be genome-wide significant in the two largest cohorts.

Response (2.1)

This is a great suggestion from the reviewer. We have added a more detailed description comparing the individual datasets (page 7) and a forest plot for visual purposes (page 7).

- 2.2** In the methods, the details given for each of the cohorts needs to be harmonized better. Some details are left out for some cohorts and not others. E.g., what software was used for the UKB? This may be what gives the methods section a disorganized feel.

Response (2.2)

Please see Response 1.6.

- 2.3** How was the conditional analysis done? Using individual level data in every cohort, or?

Response (2.3)

The conditional analysis was performed on the GWASs from Iceland, the UK and the US, separately, using true imputed genotypes of participants. The adjusted P -values were then combined for all three datasets. We have added a more detailed description of the conditional analyses to Methods (page 20).

- 2.4 Why weren't other groups in UKB assessed? What were the number of cases in those groups?

Response (2.4)

Cases of African heritage ($N = 107$) and South-Asian heritage ($N = 205$) were assessed in the UKB. Using the small datasets, we did not uncover any genome-wide signals and none of the six lead sequence variants associated with vestibular disorders (ICD-10 H81) in these two groups. Thus, we did not include the analysis in the manuscript.

SNP	Gene	EA	African				South-Asian			
			P	OR	EAF (%)	Info	P	OR	EAF (%)	Info
rs7130190	OTOG	T	0.35	1.26	12.0	0.94	0.49	0.88	9.36	0.95
rs10862089	OTOGL	T	0.38	1.29	7.02	0.98	0.17	1.22	13.5	0.98
rs428549	ZNF91	G	0.44	0.89	56.4	1.00	0.60	0.94	29.8	1.00
rs6753393	ARMC9	C	0.59	1.09	44.6	0.99	0.020	1.35	17.9	1.00
rs612969	TECTA	G	0.64	1.09	69.5	1.00	0.72	1.04	30.9	1.00
rs2272744	OTOP1	C	0.90	0.98	51.3	0.99	0.052	0.81	47.1	0.99

- 2.5 How were cryptic relatives adjusted for? It is explicitly addressed for the Icelandic data, but not mentioned for the other cohorts, e.g., no details for the UK Biobank which has a substantial amount of known relatives.

Response (2.5)

The reviewer is correct to highlight the lack of information. LD score regression was used to account for distribution inflation due to cryptic relatedness and population stratification in Iceland, the UK, and the US (page 18).

- 2.6 The data availability should cover all cohorts and be explicit as to what data is available.

Response (2.6)

The GWAS summary statistics for the vertigo meta-analysis will be available at <https://www.decode.com/summarydata/>.

Reviewer 3

AT Skuladottir and colleagues performed the first GWAS for vertigo in a large vertigo case control study (48K cases, 894.5K controls) of European ancestry and identified 6 significant loci. This is a well-executed and important study in the field.

Major comments:

- 3.1** The authors may consider calculating SNP-based / chip heritability for vertigo in the dataset.

Response (3.1)

We thank the reviewer for the suggestion. We used LD score regression to estimate the SNP heritability of vertigo in Iceland and vestibular disorders in the UK and the US. The estimated SNP heritability in Iceland is 0.12 (95% CI 0.055-0.18) and 0.23 (95% CI 0.13-0.32) in the UK. The SNP heritability in the US was not significant (page 8).

- 3.2** In addition to conventional SNP-based GWAS, the authors may complete a genome-wide gene-based association study to identify additional loci of interest (Hägg et al., Hum Mol Genet 2015).

Response (3.2)

We conducted a gene-based GWAS using MAGMA² and identified seven loci that associate with vertigo, two additional to the loci identified in the GWAS meta-analysis, at loci 5p13.3 and 6q25.1. The gene-based genome-wide association analysis has now been added to Results (page 9, Supplementary Fig. 6 and Supplementary Table 2). We also added a description of the method (page 21).

- 3.3** The authors may complete a genome-wide rare / common CNV association study for vertigo (Wheeler et al., Nat Genet 2013).

Response (3.3)

The GWASs in the Icelandic, UK, and US datasets include sequence variants and indels. Although a CNV association study may be valuable, currently we have a complete CNV calling for two of the datasets, one of which remains to be imputed. Future studies may focus on CNV associations.

- 3.4** The authors used a weighted genome-wide significance threshold based on predicted functional impact of association signals in the GWAS. They may provide a brief summary of the results for conventional GWAS for vertigo without weighted adjustment for sequence variants (equal prior probability for all variants) as a comparison.

Response (3.4)

We decided to use weighted genome-wide significance threshold based on predicted function to increase the power to detect loss-of-function and missense associations.

We have added the results for a conventional GWAS *P*-value threshold to the Results (page 5).

- 3.5** The vertigo association at *ARMC9* co-localizes with the top *cis*-eQTL for *ARMC9* ($r = 0.89$) in adipose tissue ($P = 3.3 \times 10^{-20}$, effect = 0.56 SD) in Iceland'. Adipocyte is an excellent target tissue for obesity, but I am not so sure for vertigo. It may be relevant to show additional *cis*-eQTL association data in diverse tissues (GTEx) for the SNPs.

Response (3.5)

We explored GTEx and none of the lead sequence variants were top eQTL for the corresponding gene. Additionally, we searched for associations between the lead sequence variants and *cis*-eQTL in whole blood and subcutaneous adipose tissue in the Icelandic transcriptomics dataset. The only significant association was between rs6753393 in *ARMC9* and adipose tissue. The reviewer's comment is appreciated and showing additional *cis*-eQTL associations in diverse tissues will improve the manuscript. The eQTL results have been added to Supplementary Data 3. Please see also Response 1.2.

3.6 The authors may consider adding GWAS gene enrichment studies for molecular / gene function pathways and expression in specific tissue.

Response (3.6)

We performed a gene-based enrichment analysis in FUMA, which revealed no strong enrichment for gene ontologies. Thus, we did not include the analysis in the manuscript. The top enrichment gene sets are shown in the table below.

Gene set	Genes	P	β
GO_SENSORY_PERCEPTION	OTOGL, OTOG, TECTA	0.060	0.59
GO_SENSORY_PERCEPTION_OF_MECHANICAL_STIMULUS	OTOGL, OTOG, TECTA	0.060	0.59
GO_NERVOUS_SYSTEM_PROCESS	OTOGL, OTOG, TECTA	0.060	0.59

- 3.7 The authors may estimate the % of variance for the trait explained by the 6 variants in an independent dataset.

Response (3.7)

We constructed a genetic risk score and estimated the increase in variance explained to be 0.26%. The results can be seen on page 8 and the method can be seen on page 20.

- 3.8 The authors may check whether the six GWAS lead SNPs or SNPs in strong linkage disequilibrium are associated with other traits / diseases in the GWAS catalogue (Kaur et al., *Obes Rev* 2019).

Response (3.8)

We appreciate the reviewer's suggestion. None of the lead SNPs associate with other traits in the GWAS catalog (<https://www.ebi.ac.uk/gwas/home>). We calculated the genetic correlation between the vertigo meta-analysis and 600 traits reported by Watanabe³. The strongest genetic correlation was with pain traits (page 8, Supplementary Fig. 5, Supplementary Data 4). We have added the description of the genetic correlation analyses in Methods (page 20).

- 3.9 None of the variants identified in the meta-analysis associate with hearing loss and only one has a modest effect on ARHI (rs612969-G in *TECTA*; $P = 9.9 \times 10^{-5}$, OR = 1.02), which may indicate a different pathology, in part, in these two systems'. I do not agree at all with this statement. This is inconsistent with the fact that rare variants in three of the six genes identified in the meta-analysis are reported to cause hearing loss. Statistical power may explain the divergent findings for SNPs. Did you consider screening the three remaining GWAS candidate genes for rare variants associated with hearing loss?

Response (3.9)

Using substantial datasets of age-related hearing impairment (ARHI; 121,934 cases and 591,699 controls), conductive and sensorineural hearing loss (ICD-10 code H90; 22,879 cases and 926,687 controls) and other hearing loss (ICD-10 code H91; 20,845 cases and 923,892 controls), we searched for rare and common variants in the six candidate GWAS genes associating with each trait. The number of variants tested in the ARHI dataset was 23,217 and 39,040 in the other two datasets. None of the variants associated significantly with the traits.

In agreement with the reviewer, these results are inconsistent with previous reports and may be explained by statistical power as previous reports are case studies and this current meta-analysis is a large study of four populations.

We have added

“... none of the variants identified in the current meta-analysis associate with hearing loss, identified by ICD-10 codes H90 and H91 (Supplementary Table 3), **nor any rare variants in the six GWAS candidate genes.**” (page 9)

and

“Rare variants in the six GWAS candidate genes do not associate with ARHI.”
(page 9)

We have also removed the rather bold suggestion “...which may indicate a different pathology, in part, in these two systems”.

3.10 The authors may consider adding a strengths / limitations section in the discussion section.

Response (3.10)

The reviewer is correct in pointing this out. A limitation section has been added to the Discussion (page 13).

Minor comments:

3.11 Please mention in the title and abstract that the study has been performed in participants of European ancestry.

Response (3.11)

We thank the reviewer for the suggestion and have added “...in individuals of European ancestry” to the abstract (page 2).

3.12 Please define what the abbreviation BBPV means.

Response (3.12)

In the Introduction (page 3), the abbreviation BPPV is introduced as benign paroxysmal positional vertigo: “Peripheral causes include benign paroxysmal positional vertigo (BPPV)...”.

3.13 Figure 1: please provide more explanation for the ‘support for gene’ section

Response (3.13)

A more specific explanation has been added for the ‘Support for gene’ section to Fig. 1. Legend (page 6).

We look forward to hearing from you in due time regarding our submission and to respond to any further questions and comments you may have.

References

1. Privé, F., Luu, K., Blum, M. G. B., McGrath, J. J. & Vilhjálmsón, B. J. Efficient toolkit implementing best practices for principal component analysis of population genetic data. *Bioinformatics* **36**, 4449–4457 (2020).
2. de Leeuw, C. A., Mooij, J. M., Heskes, T. & Posthuma, D. MAGMA: Generalized Gene-Set Analysis of GWAS Data. *PLoS Comput. Biol.* **11**, (2015).
3. Watanabe, K. *et al.* A global overview of pleiotropy and genetic architecture in complex traits. *Nat. Genet.* **51**, 1339–1348 (2019).

REVIEWERS' COMMENTS:

Reviewer #2 (Remarks to the Author):

The authors have responded well to my questions.

Reviewer #3 (Remarks to the Author):

The authors adressed adequately all my comments, thank you.